# The use of a chimeric antigen for *Plasmodium falciparum* and *P. vivax* seroprevalence estimates from community surveys in Ethiopia and Costa Rica

**Jessica N. McCaffery**[1], **Balwan Singh**[2], **Douglas Nace**[2], **Ashenafi Assefa**[3,4], **Jimee Hwang**[2,5], **Mateusz Plucinski**[2,5], **Nidia Calvo**[6], **Alberto Moreno**[1], **Venkatachalam Udhayakumar**[2], **Eric Rogier**[2]*

**1** Emory Vaccine Center, Emory National Primate Research Center, Emory University, Atlanta, Georgia, United States of America, **2** Malaria Branch, Division of Parasitic Diseases and Malaria, Centers for Disease Control and Prevention, Atlanta, Georgia, United States of America, **3** Ethiopian Public Health Institute, Addis Ababa, Ethiopia, **4** School of Public Health, Addis Ababa University, Addis Ababa, Ethiopia, **5** U.S. President's Malaria Initiative, Atlanta, Georgia, United States of America, **6** Instituto Costarricense de Investigación y Enseñanza en Nutrición y Salud (INCIENSA), Tres Rios, Costa Rica

* jmccaffery@cdc.gov

## Abstract

### Background

In low-transmission settings, accurate estimates of malaria transmission are needed to inform elimination targets. Detection of antimalarial antibodies provides exposure history, but previous studies have mainly relied on species-specific antigens. The use of chimeric antigens that include epitopes from multiple species of malaria parasites in population-based serological surveys could provide data for exposure to multiple *Plasmodium* species circulating in an area. Here, the utility of *P. vivax/P. falciparum* chimeric antigen for assessing serological responses was evaluated in Ethiopia, an endemic country for all four human malarias, and Costa Rica, where *P. falciparum* has been eliminated with reports of sporadic *P. vivax* cases.

### Methods

A multiplex bead-based assay was used to determine the seroprevalence of IgG antibodies against a chimeric malaria antigen (PvRMC-MSP1) from blood samples collected from household surveys in Ethiopia in 2015 (n = 7,077) and Costa Rica in 2015 (n = 851). Targets specific for *P. falciparum* (PfMSP1) and *P. vivax* (PvMSP1) were also included in the serological panel. Seroprevalence in the population and seroconversion rates were compared among the three IgG targets.

### Results

Seroprevalence in Costa Rica was 3.6% for PfMSP1, 41.5% for PvMSP1 and 46.7% for PvRMC-MSP1. In Ethiopia, seroprevalence was 27.6% for PfMSP1, 21.4% for PvMSP1,

**Data Availability Statement:** All relevant data are within the paper and its Supporting information files.

**Funding:** The authors received no specific funding for this work.

**Competing interests:** The authors have declared that no competing interests exist.

and 32.6% for PvRMC-MSP1. IgG levels in seropositive individuals were consistently higher for PvRMC-MSP1 when compared to PvMSP1 in both studies. Seroconversion rates were 0.023 for PvMSP1 and 0.03 for PvRMC-MSP1 in Costa Rica. In Ethiopia, seroconversion rates were 0.050 for PfMSP1, 0.044 for PvMSP1 and 0.106 for PvRMC-MSP1.

## Conclusions

Our data indicate that chimeric antigen PvRMC-MSP1 is able to capture antibodies to multiple epitopes from both prior *P. falciparum* and *P. vivax* infections, and suitable chimeric antigens can be considered for use in serosurveys with appropriate validation.

## Introduction

In 2021, the World Health Organization reviewed the progress made towards malaria elimination since 2000 and placed increased emphasis on examining the global trends in the burden of malaria to identify any variations in malaria burden by age as malaria transmission declines [1]. Reliable measures of transmission are particularly critical in the setting of malaria elimination as decreasing malaria incidence is associated with an increase in the proportion of asymptomatic infections as well as a decrease in treatment-seeking behavior and identified cases for epidemiological estimates [2–10]. Therefore, conventional diagnostic methods which provide data on active infection, such as microscopy or malaria rapid diagnostic tests, are not as well suited for estimating the true level of recent or ongoing malaria exposure in a population.

In recent years, serological surveillance methods have repeatedly demonstrated utility in measuring transmission in low malaria-endemic regions [2–10]. Serological methods offer multiple advantages well suited for use in surveys, primarily their ability to generate estimates of malaria exposure history in a population by detecting antibodies against parasite antigens [11,12]. Antibody seroprevalence and the age of participants are used to estimate the rate of antimalarial antibody acquisition in a population [9,11]. These estimates can be used to monitor differences among populations in an area, or changes in transmission over time, or in response to an intervention [7].

While serological methods have seen increased use in the assessment of malaria transmission, these studies have mainly relied on recombinant antigens based on a single antigen or on epitopes that share sequence identity with naturally occurring *Plasmodium* antigens [13]. Multiplex assays allow for the simultaneous assessment of antibody responses to multiple antigens from a single sample [12,14]. It remains to be determined if chimeric antigenic constructs from different malaria parasite species can be used to collect serology data for multiple species in serosurveys.

A recent study from our laboratory described the pattern of serological response in U.S. travelers [15] using a chimeric protein based on *Plasmodium vivax* MSP1 and *P. falciparum* circumsporozoite protein (CSP) repeat epitopes [16], known as PvRMC-MSP1. It was evident from this study that this chimeric antigen captured IgG from a majority of returning U.S. travelers with PCR confirmed malaria infection regardless of the *Plasmodium* species responsible for infection [15]. Furthermore, an increased assay signal was observed for PvRMC-MSP1 compared to recombinant PvMSP1 in 34 out of 38 active *P. vivax* infections [15].

A better understanding of the utility of chimeric antigens for capturing antibody responses in endemic populations with different levels of malaria exposure will help determine future applications of such tools for malaria serological studies. Therefore, this study evaluated the

ability of PvRMC-MSP1 to capture IgG from dried blood spot samples (DBS) collected from Ethiopia and Costa Rica. Ethiopia was selected for this study because the country is co-endemic for both *P. falciparum* and *P. vivax*, making up 69% and 27% of malaria cases, respectively [17]. Costa Rica was selected as a representative low, mono-species *Plasmodium* endemic region, and as of 2017, 98% of the country was considered malaria-free, with only three remaining foci of active transmission for *P. vivax* [18]. A multiplex bead-based assay was used to evaluate the population-level IgG binding to PvRMC-MSP1 in comparison to recombinant *P. falciparum* MSP1 and recombinant *P. vivax* MSP1 via modeled seroprevalence curves and LOESS regression curves.

## Materials and methods

### Study sites, design, enrollment, and ethics statements

Ethiopian samples were collected as part of the 2015 Ethiopia Malaria Indicator Survey (MIS) [19], a national household survey that occurred between September 30 and December 10, 2015, which coincided with the high malaria transmission season and covered 13,875 house-holds in malarious areas between 2,000m and 2,500m above sea level. Two-stage cluster ran-domized sampling was used to select 555 areas and 25 households within each area to ensure that the survey participants were representative of malaria endemic regions throughout the country. In each household, blood samples were obtained from all children under five years of age upon obtaining informed consent from their parent/guardian. In Ethiopia, the informed consent forms and questionnaires were translated into Amharic, Oromiffa, and Tigrigna lan-guages and read aloud in their entirety by the survey team in the participant's language of choice to ensure understanding before verbal consent was obtained from the adult participant, the parent/guardian of any participant below the age of 18, and verbal assent for children below the age of 18. Verbal consent/assent was documented along with responses to the ques-tionnaire. In every fourth household, persons of all ages were enrolled in the study upon obtaining informed consent. Blood was spotted onto filter paper and allowed to dry (creating dried blood spots, DBS) before each sample was individually packaged. The MIS-2015 protocol received ethical clearance from Ethiopia's National Research Ethics Review Committee. The survey protocol underwent human subjects review at the US Centers for Disease Control and Prevention (CDC) and received non-research determination. Additional ethical clearance for the present serology study was obtained from the Institutional Review Board of the College of Health Sciences of the Addis Ababa, University (AAUMF 03–008). From the 2015 Ethiopia MIS, 7,077 DBS were available for serological data collection.

In the Costa Rican canton of Matina in 2015, a total of 851 individuals were enrolled in their households and provided blood samples for DBS creation. The canton of Matina in the province of Limón was selected for this study, given that it was one of the last locations in the country where malaria cases were present. In Costa Rica, informed consent forms were read aloud in Spanish by the survey team before signatures were obtained from participating adults older than 18, parental/guardian consent for any participant below the age of 18, and assent from children 12–17. At enrollment, a questionnaire was administered and whole blood sam-ples were collected on filter paper for molecular and serological analysis. The study was approved by the Costa Rica Ministry of Health (DVS-721-2015), and the study protocol under-went human subjects review at CDC and received non-research determination.

For both Ethiopian and Costa Rican studies, the laboratory staff at the CDC and Emory University had no contact with the participants of this study and no access to any personal identifiers.

## Recombinant and chimeric *Plasmodium* antigens used for multiplex assay

The structure and production of the chimeric *P. vivax* merozoite surface protein 1 has been previously described [15,16]. Briefly, the *P. vivax* recombinant modular chimera based on MSP1 (PvRMC-MSP1) was based on the *P. vivax* Belem sequence (GenBank: XP_001614842.1) and contains five promiscuous T cell epitopes, with two also functioning as B cell epitopes, as well as an extended version of the *P. vivax* MSP1 19 kDa fragment containing two T helper epitopes from the MSP1 33 kDa fragment, and the $(NANP)_6$ repeat peptide from *P. falciparum* circumsporozoite protein (CSP) [16]. Each T cell epitope included in PvRMC-MSP1 is 19 amino acids in length. In addition to PvRMC-MSP1, recombinant PfMSP1 and PvMSP1 antigens and the NANP repeat peptide from PfCSP were also used; the production and use of these antigens has been reported previously [8,20].

## Multiplex bead-based assay (MBA)

Magnetic MagPlex microspheres (Luminex Corp.) were covalently linked to the chimeric PvRMC-MSP1 protein, the recombinant PfMSP1 and PvMSP1 antigens, and the PfCSP peptide as previously described [12,15], with all three recombinant proteins conjugated at a concentration of 20 μg/mL and the peptide at a concentration of 30 μg/mL. To obtain whole blood from the filter paper, a 6 mm diameter hole punch from individual DBS samples were eluted overnight at 4˚C in buffer containing: PBS, 0.05% Tween 20, 0.5% polyvinylpyrrolidone, 0.5% poly(vinyl) alcohol, 0.1% casein, 0.5% BSA, 0.02% $NaN_3$, and 3 μg/mL *E. coli* extract to prevent nonspecific binding. Samples were subsequently diluted to 1:100 serum concentration for the immunoassay.

For the multiplex immunoassay, all data were collected with an overnight incubation assay as described previously [14]. Briefly, in 5 mL of reagent buffer (PBS, 0.05% Tween 20, 0.5% BSA, 0.02% $NaN_3$), a bead master mix was prepared with all conjugated beads included, and 50 μL of the bead mix was pipetted into each well of a BioPlex Pro plate (Bio-Rad, Hercules, CA). Beads were washed twice on a handheld magnet (Luminex Corp, Austin, TX) with 100 μL wash buffer (PBS, 0.05% Tween 20). After washing, 50 μL of reagent mix (in 5 mL of reagent buffer: 1:500 anti-human IgG, 1:625 anti-human IgG4, 1:200 streptavidin-PE) was added to all wells, then 50 μL samples (or controls) were added to the reagent mix in the appropriate wells. Anti-IgG4 was added separately to the anti-human IgG mixture as we have previously found that anti-IgG4 responses are difficult to determine using only the standard anti-IgG cocktail [21]. All anti-human antibodies were obtained from Southern Biotech (Birmingham, AL) and streptavidin-PE from Invitrogen (Waltham, MA). Plates were incubated overnight with gentle shaking at room temperature and protected from light. The following day (after ~ 16 h total incubation time), plates were washed three times, and beads were resuspended with 100 μL PBS and read on a MAGPIX machine (Luminex Corp). MFI signal was generated for a target of 50 beads/region. A background MFI from wells including only elution buffer was subtracted from each sample to give a final value of MFI-bg.

## Statistical analyses

A seropositivity threshold was generated for each antigen by assaying a panel of 92 U.S. resident blood donors without recent travel outside the country and determining the lognormal mean plus three standard deviations for the MFI-bg signal. A reversible catalytic model was fit to the seropositivity by age data for each antigen, and the estimates for the serological conversion rate (SCR) and serological reversion rate (SRR) per year were directly calculated from the likelihood model [22] utilizing R version 3.3.2 (R Foundation for Statistical Computing, Vienna, Austria). Local regression (LOESS) curves for IgG levels by age were generated in

SASv9.2 (Cary, NC) by the sgplot procedure with cubic interpolation and a smoothing degree of 1.

## Results

### Ethiopia and Costa Rica study populations

From the 2015 Ethiopia MIS, 7,077 DBS were available for serological data collection, and 851 DBS were available from the 2015 Costa Rica survey. In the Ethiopia survey, 53.1% of participants with serological data were male, and nearly half (47.2%) of persons were five years of age or younger (Table 1). From the Costa Rica survey, 32.5% of participants with serological data were male, and the youngest participant was 12 years old.

### Population-level response to antigens in Ethiopia and Costa Rica

The percentage of participants from each survey who were IgG positive (seropositive) for each of the markers varied substantially by antigen and study population. Overall seroprevalence in Costa Rica was 3.6% for PfMSP1, 41.5% for PvMSP1 and 46.7% for PvRMC-MSP1. In Ethiopia, overall seroprevalence was 27.6% for PfMSP1, 21.4% for PvMSP1 and 32.6% for PvRMC-MSP1. For both surveys, increases in both seropositivity and IgG levels were noted with an increase in age (Figs 1 and 2), though these increases were minor for the PfMSP1 antigen in the Costa Rican population. For Ethiopia specifically, plotting the IgG assay signal against age revealed that antibodies against the PfMSP-1 antigen were highest within the Ethiopian populace, followed by PvRMC-MSP1, and lastly, PvMSP-1 (Fig 1B). A different pattern was noted in the Costa Rican populace, with the PvRMC-MSP1 antigen providing the highest IgG signal among all ages, followed by IgG against PvMSP1 and very low IgG levels against PfMSP1 for all ages (Fig 2B).

### Modeling for seroconversion rates

For the Ethiopian data, the highest seroconversion rate was observed for PvRMC-MSP1 at 0.106 (95% CI: 0.095–0.116), compared to PfMSP1 at 0.050 (95% CI: 0.045–0.055) and PvMSP1 at 0.044 (95% CI: 0.039–0.049) (Table 2). Analysis of the Costa Rican data revealed the seroconversion rate of PvRMC-MSP1 to be the highest of the three antigens assayed, at

**Table 1. Demographic characteristics of the Costa Rica and Ethiopia study populations.**

| Study, Variable | Number (%) |
|---|---|
| **Ethiopia** | |
| Male | 3757 (53.1%) |
| Female | 3320 (46.9%) |
| Age 0-5y | 3342 (47.2%) |
| Age 6-10y | 774 (10.9%) |
| Age 11-20y | 933 (13.2%) |
| Age 21-30y | 749 (10.6%) |
| Age >30y | 1279 (18.1%) |
| **Costa Rica** | |
| Male | 277 (32.5%) |
| Female | 574 (67.5%) |
| Age 0-5y | 0 (0.0%) |
| Age 6-10y | 0 (0.0%) |
| Age 11-20y | 119 (14.0%) |
| Age 21-30y | 139 (16.3%) |
| Age >30y | 593 (69.7%) |

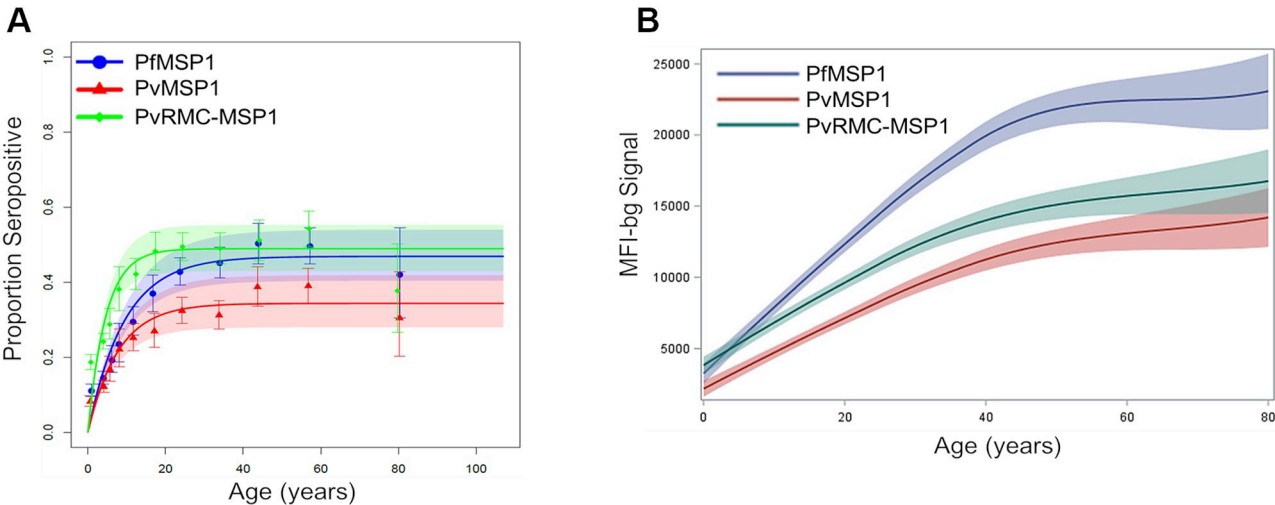

**Fig 1. Population-level responses to PvRMC-MSP1, PfMSP1, and PvMSP1 from Ethiopian persons.** A) Modelled seroprevalence curves for the Ethiopian study population for the three antigens, with seroconversion estimates shown in Table 2. B) LOESS regression curves for IgG assay signal of the study population to each antigen by age. A minimal anonymized data set for the responses to PvRMC-MSP1 in Ethiopia is included with the Supporting Information for this publication.

0.030 (95% CI: 0.02–0.04), followed by PvMSP1 at 0.023 (95% CI: 0.02–0.03), and then PfMSP1 at 0.00086 (95% CI: -0.001–0.003).

## Assay signal between PvRMC-MSP1 and PfCSP

Although PvRMC-MSP1 is based primarily on the *P. vivax* Belem sequence, six copies of the NANP repeat sequence from the *P. falciparum* circumsporozoite protein (PfCSP) are present at the C-terminus of PvRMC-MSP1 and were included during the design of PvRMC-MSP1 to

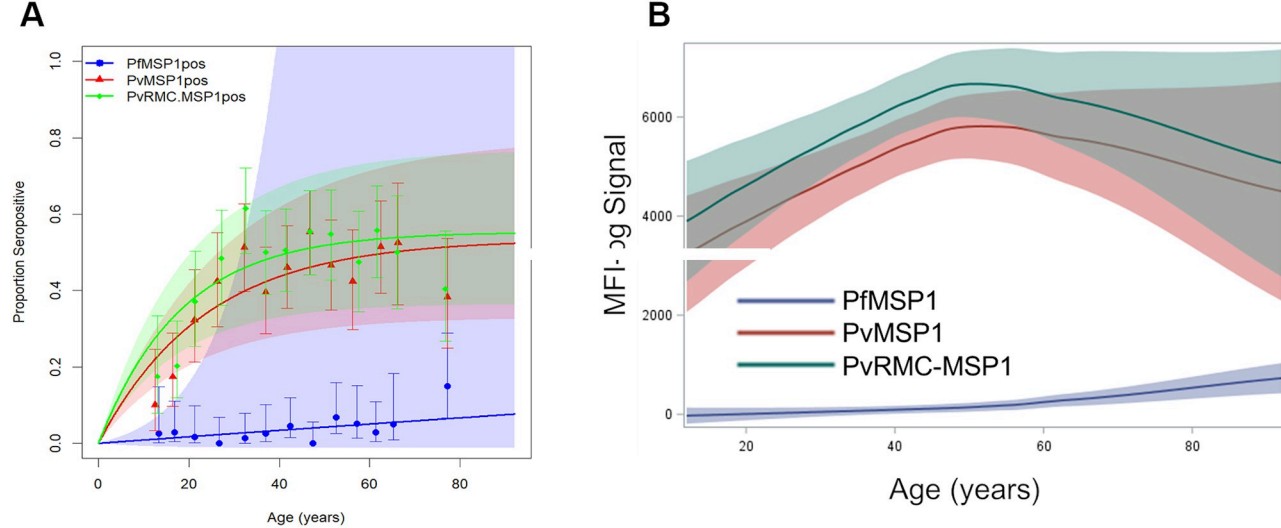

**Fig 2. Population-level responses to PvRMC-MSP1, PfMSP1, and PvMSP1 from Costa Rican persons.** A) Modelled seroprevalence curves for the Costa Rican study population for the three antigens, with seroconversion estimates shown in Table 2. B) LOESS regression curves for IgG assay signal of the study population to each antigen by age. The x-axis for the LOESS regression curves starts at age 12, corresponding to the youngest participant from the Costa Rican survey. A minimal anonymized data set for the responses to PvRMC-MSP1 in Costa Rica is included with the Supporting Information for this publication.

**Table 2. Serological conversion rate for plasmodium antigens in the Ethiopian and Costa Rican study populations.**

| Country | Antigen | Point Estimate | Lower 95% bound | Upper 95% bound |
|---|---|---|---|---|
| **Ethiopia** | | | | |
| | PfMSP1 | 0.05 | 0.045 | 0.055 |
| | PvMSP1 | 0.044 | 0.039 | 0.049 |
| | PvRMC-MSP1 | 0.106 | 0.095 | 0.116 |
| **Costa Rica** | | | | |
| | PfMSP1 | 0.00086 | -0.001 | 0.003 |
| | PvMSP1 | 0.023 | 0.02 | 0.03 |
| | PvRMC-MSP1 | 0.03 | 0.02 | 0.04 |

serve as an additional purification tag. To determine if antibody recognition and binding to the PfCSP region is responsible in part for the assay signals observed for PvRMC-MSP1, the MFI-bg signals were compared between the chimera and a peptide containing five copies of the PfCSP NANP peptide (NANPx5, S1 Fig). For the Ethiopian study population, a majority of samples produced higher MFI-bg signals in response to antigen capture by PvRMC-MSP1 than for PfCSP NANPx5 peptide, with a comparatively smaller subset of samples tracking along the x = y reference line or showing increased signal for PfCSP than PvRMC-MSP1 (S1A Fig). The MFI-bg signal obtained for the PvRMC-MSP1 and PfCSP antigens from individuals from Costa Rica was also compared, revealing primary binding to PvRMC-MSP1 as compared to PfCSP, with relatively few samples following the x = y reference line (S1B Fig).

## Discussion

Malaria serological data offers the advantage of expanding the window of time to assess prior malaria infection in the human population because exposed individuals maintain anti-malaria IgG antibodies for months to years after exposure to malaria parasites [22], and this methodology can use targets that are not affected by short-term variations in malaria transmission [5]. Additionally, these methods can detect antibodies from convenient samples, such as DBS, which are easier to collect and store than blood fractionation techniques and cold storage [23]. Assessing the presence of antibodies for multiple antigen targets through bead-based multiplex assay makes serological studies high throughput and cost-effective for application in seroepidemiologic studies aimed at understanding the burden of malaria transmission within a community [9,11]. As a result, these characteristics make malaria serological studies an excellent tool for evaluating prevalence in malaria elimination settings.

Although there are still few reports of malaria seroprevalence studies using designed antigens, the potential of a chimeric antigen approach has been recently demonstrated in a study assessing malaria transmission intensity along the China-Myanmar border [13]. This study used a designed antigen that included 11 epitopes from eight *P. falciparum* antigens, including single epitopes from CSP and MSP2, and two epitopes each from AMA1 and MSP1, which was able to differentiate individuals living in endemic and non-endemic regions within the study area [13]. However, unlike the PvRMC-MSP1 antigen reported here, the design of the antigen used in the China-Myanmar study aimed to avoid cross-reactivity with *P. vivax*, highlighting the potential to customize engineered seroprevalence antigens for a seroprevalence study to either include all *Plasmodium* species or target only one. However, the chimeric antigen PvRMC-MSP1 used here was originally designed as an experimental vaccine construct that included promiscuous T cell epitopes from *P. vivax* Belem strain, *P. vivax* MSP-1 19kDa antigen and six repeats of NANP sequence from the CSP antigen of *P. falciparum* [15,16,24].

Therefore, the use of the PvRMC-MSP1 antigen allowed for the investigation of how a hybrid recombinant antigen with epitopes from two different malaria parasites could capture antibody reactivity patterns from endemic populations with varying history of transmission pattern and exposure history.

Two different malaria-endemic settings were utilized for this study. Ethiopia is a country in the horn of Africa that is known to be endemic for all four human malarias [5] and is unique in the African context in that *P. vivax* accounts for approximately one-third of reported clinical malaria cases [17]. Ethiopia has seen reductions in its malaria burden and aims for malaria elimination by 2030. Costa Rica is a country in Central America that has drastically suppressed malaria transmission to the point where nearly all identified cases are identified as being imported infections [25]. Additionally, though *P. falciparum* was noted in Costa Rica many decades ago [26], most recent cases are caused by *P. vivax* [18]. The malaria transmission settings for these two countries were reflected in both the seroprevalence to the antigen targets used in the multiplex assay as well as the estimated seroconversion rates.

For the species-specific antigens in the Ethiopia study, more PfMSP1 seropositive individuals were identified when compared with PvMSP1, with 28% and 21% of study participants seropositive for these two targets, respectively. The chimeric PvRMC-MSP1 antigen provided even more seropositive calls, with 33% of all blood samples having IgG against this target. This increase in the seroprevalence estimate is likely due to the cumulative capture of antibodies to multiple epitopes such as universal T cell epitopes of PvMSP1 and PfCSP epitopes, in addition to PvMSP1 19kDa antigen (S1A and S1B Fig) [15,16,24]. Though the PvRMC-MSP1 antigen provided the highest seroprevalence estimate in the Ethiopian population, the assessment of the IgG levels found that by far the PfMSP1 antigen had the highest titers in the population, which may be due to the increased transmission as well as multiple exposures of the Ethiopian study population to *P. falciparum*.

In the Costa Rican setting, seropositivity to PfMSP1 was almost nonexistent, with only the oldest individuals showing any notable levels (approximately 10% seropositivity in persons over 70 years of age, versus 2.9% in persons younger than 70). The minimal response to PfMSP1 was not a surprising finding because *P. falciparum* has rarely been reported in Costa Rica over the past fifty years [25,26]. Seroprevalence was 42% for PvMSP1 and 47% for PvPMC-MSP1, consistent with our hypothesis that this reactivity pattern can be attributed to cumulative response due to the presence of multiple epitopes of *P. vivax* MSP1 besides the 19kDa domain in the chimeric antigen.

Estimates for seroconversion rates for the three antigens in the Ethiopian and Costa Rican studies provided further information for the transmission dynamics of malaria parasites in these populations over time and displayed the utility of the chimeric antigen for malaria serological estimates. In Ethiopia, the increased seroconversion rate for PvRMC-MSP1 over the PfMSP1 and PvMSP1 antigens (0.11 versus 0.05 and 0.04 respectively) showed the increased rate at which antibodies that recognize this antigen are acquired in children compared to the other MSP1 antigens. Indeed, the 95% confidence interval for the regression curve fitting for the chimeric antigen was completely separated from the confidence intervals for the other two antigens in the 0-15y age group (Fig 1A), and the overall estimated seroconversion rate was statistically higher for the chimera (Table 2). The population considered seropositive plateaued for all three antigens around age 30, though average IgG levels continued to increase as persons aged in the population (Fig 1A versus 1B). The IgG signal for PvRMC-MSP1 is primarily due to anti-*P. vivax* antibodies, but since the antigen can also capture anti-PfCSP antibodies, these seroprevalence and seroconversion data should be interpreted as exposure to either of these two malaria parasites.

Assessment of seroconversion rates in Costa Rica provided further evidence of the increased sensitivity of PvRMC-MSP1 over PvMSP1 for detecting *P. vivax* exposure in this low-endemic setting. The seroconversion rate for PvRMC-MSP1 for the study population from the canton of Matina in Costa Rica was again the highest of the three antigens tested. However, all three of these rates were more than an order of magnitude lower in Costa Rica when compared to the Ethiopian estimates (Table 2), alluding to the much lower exposure of the Costa Rican population to malaria parasites over the past decades when compared with the Ethiopian populace. Two limitations to the Costa Rica study included enrollment of persons only aged 12 years and older and the relatively fewer numbers (n = 851) of persons enrolled. For these reasons and the very low overall incidence of malaria transmission, modeling for seroconversion rates showed wider and overlapping confidence intervals with both PvRMC-MSP1 and PvMSP1. Not surprisingly, seropositivity to PfMSP1 was almost nonexistent except in the oldest persons who may have been exposed decades ago [18,26,27].

Some limitations of this study include a) The chimeric construct used here was designed to elicit broad T cell reactivity as a vaccine candidate and contained only selected antigenic domains. It will be useful to compare different chimeric constructs with broader inclusion of B cell epitopes from multiple antigens in future studies; b) The study population was limited to two different endemic populations with very different transmission patterns. Future studies that include specimens from multiple other endemic settings (including diverse age groups) will provide additional insights for assessing the value of chimeric antigens. Despite these limitations, the increased sensitivity of PvRMC-MSP1 to capture antimalarial IgG even in younger populations and in a low endemic setting highlights the potential utility of well-designed chimeric antigens for use in seroepidemiological studies. Chimeric antigens can be constructed using recombinant technology as applied in the development PvRMC-MSP1 construct or synthesized using longer multiple antigen peptides with defined epitopes from different species or target antigens [28]. Exploring such chimeric antigen constructs can broaden the toolbox for seroepidemiological studies and may help to improve their utility in populations with limited ongoing exposure.

## Conclusion

The data from this study demonstrate the potential use of chimeric antigens for population-level serosurveillance as these antigens can be tailored to capture antibodies to multiple epitopes in different endemic settings/populations. Engineered antigens based on highly recognized epitopes may also allow for increased sensitivity due to the reorganization of the antigen to expose these highly recognized epitopes while simultaneously removing sequences of low antigenic value. Taken together, the use of engineered antigens and the bead-based multiplex technique can allow for robust data collection for seroepidemiological studies, especially for malaria elimination campaigns.

## Supporting information

**S1 Fig. Cross-binding of anti-PfCSP IgG with PvRMC-MSP1.** A) Scatterplot of PfCSP signal compared to PvRMC-MSP1 for the Ethiopian study population. B) Scatterplot of PfCSP signal compared to PvRMC-MSP1 for Costa Rican study population. The hashed reference line shown is y = x.
(TIF)

**S1 Dataset.**
(XLS)

**S2 Dataset.**
(XLS)

## Acknowledgments

The authors would like to acknowledge all the field teams and survey participants from these studies.

## Disclaimer

The findings and conclusions in this report are those of the authors and do not necessarily represent the official position of the Centers for Disease Control and Prevention.

## Author Contributions

**Conceptualization:** Ashenafi Assefa, Nidia Calvo, Eric Rogier.

**Data curation:** Jessica N. McCaffery, Balwan Singh, Douglas Nace, Ashenafi Assefa, Eric Rogier.

**Formal analysis:** Jessica N. McCaffery, Ashenafi Assefa, Mateusz Plucinski, Nidia Calvo, Alberto Moreno, Venkatachalam Udhayakumar, Eric Rogier.

**Investigation:** Balwan Singh, Douglas Nace, Ashenafi Assefa, Eric Rogier.

**Methodology:** Balwan Singh, Douglas Nace, Ashenafi Assefa, Eric Rogier.

**Project administration:** Eric Rogier.

**Resources:** Ashenafi Assefa, Jimee Hwang, Mateusz Plucinski, Nidia Calvo, Alberto Moreno, Venkatachalam Udhayakumar, Eric Rogier.

**Software:** Jimee Hwang, Mateusz Plucinski, Venkatachalam Udhayakumar.

**Supervision:** Venkatachalam Udhayakumar.

**Visualization:** Jessica N. McCaffery, Ashenafi Assefa, Mateusz Plucinski.

**Writing – original draft:** Jessica N. McCaffery.

**Writing – review & editing:** Jessica N. McCaffery, Eric Rogier.

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
