## [Decision Letter · Decision Letter 0]

2 Mar 2022

PONE-D-22-01720The use of a chimeric antigen for Plasmodium falciparum and P. vivax seroprevalence estimates from community surveys in Ethiopia and Costa RicaPLOS ONE

Dear Dr. McCaffery,

Thank you for submitting your manuscript to PLoS ONE. After careful consideration, we feel that your manuscript will likely be suitable for publication if the authors revise it to address specific points raised by the reviewers. According to the reviewers, there are some specific areas where further improvements would be of substantial benefit to the readers.   For your guidance, a copy of the reviewers' comments was included below. 

We look forward to receiving your revised manuscript.

Kind regards,

Luzia Helena Carvalho, Ph.D.

Academic Editor

PLOS ONE

Journal Requirements:

3. You indicated that you had ethical approval for your study. In your Methods section, please ensure you have also stated whether you obtained consent from parents or guardians of the minors included in the study or whether the research ethics committee or IRB specifically waived the need for their consent.

4. Please provide additional details regarding participant consent. In the ethics statement in the Methods and online submission information, please ensure that you have specified what type you obtained (for instance, written or verbal, and if verbal, how it was documented and witnessed). If your study included minors, state whether you obtained consent from parents or guardians. If the need for consent was waived by the ethics committee, please include this information.

Reviewers' comments:

Reviewer's Responses to Questions

**Comments to the Author**

1. Is the manuscript technically sound, and do the data support the conclusions?

Reviewer #1: Yes

2. Has the statistical analysis been performed appropriately and rigorously? 

Reviewer #1: Yes

3. Have the authors made all data underlying the findings in their manuscript fully available?

Reviewer #1: Yes

4. Is the manuscript presented in an intelligible fashion and written in standard English?

Reviewer #1: Yes

5. Review Comments to the Author

Reviewer #1: Throughout the article the acronym “PvRMC-MSP1” was used a few times for the chimeric malaria antigen however, it would be better to use the term (PvRMC-MSP1/PfCSP) on the whole text, as the use of the first term could lead to a misinterpretation, that the antigen used had only specific targets for vivax.

In the introduction, it would be better to use the most current reference from the World Health Organization – Malaria Report 2021 (References 1, 17, 18).

In the “Materials and Methods - Study sites, design, enrollment, and ethics statements” in relation to Ethiopia, briefly explain how the areas of study were selected. Insert in this topic the total number of participants from Ethiopia.

Explain - From the Costa Rica the youngest participant was 12 years old however, in Figure 2b, there are data for children under 12 years old.

In the discussion, in relation to Ethiopia, it would be recommended comparing the data from the different selected areas of the country relating to seroconversion rate and the prevalence of the different species of plasmodium. Use tables and figures to relate this data.

6. PLOS authors have the option to publish the peer review history of their article (what does this mean?). If published, this will include your full peer review and any attached files.

Reviewer #1: No

---

## [Author Response · Author response to Decision Letter 0]

11 Mar 2022

Response to Reviewer Comments

Manuscript Title: The use of a chimeric antigen for Plasmodium falciparum and P. vivax seroprevalence estimates from community surveys in Ethiopia and Costa Rica

PLOS ONE Submission ID: PONE-D-22-01720

Reviewer 1 - Review Comments to the Author:

1. Throughout the article the acronym “PvRMC-MSP1” was used a few times for the chimeric malaria antigen however, it would be better to use the term (PvRMC-MSP1/PfCSP) on the whole text, as the use of the first term could lead to a misinterpretation, that the antigen used had only specific targets for vivax.

We thank the reviewer for this suggestion, but this has previously been the name designated to this antigen by our multiple previous publications, and it should remain the same for consistency. Throughout the manuscript, we state multiple times that PvRMC-MSP1 also contains the epitopes for PfCSP, so this is made clear throughout the whole text for the reader’s interpretation.

2. In the introduction, it would be better to use the most current reference from the World Health Organization – Malaria Report 2021 (References 1, 17, 18).

The reviewer is correct that newer references should be used where possible. The World Health Organization Malaria Report citation has been updated to the 2021 version for references 1. The first sentence has also been updated to “In 2021, the World Health Organization reviewed the progress made towards malaria elimination since 2000 and placed increased emphasis on examining the global trends in the burden of malaria to identify any variations in malaria burden by age as malaria transmission declines (1).” to reflect the focus of the most recent WHO Malaria report.

The 2018 Malaria Country Profiles for Ethiopia and Costa Rica are the most recent versions of these reports available for these countries from the WHO.

3. In the “Materials and Methods - Study sites, design, enrollment, and ethics statements” in relation to Ethiopia, briefly explain how the areas of study were selected. Insert in this topic the total number of participants from Ethiopia.

The household survey samples obtained from the multiple sites in Ethiopia correspond to the sites included in the 2015 Multiple Indicator Survey. The study was conducted from September 30, to December 10, 2015, covering a sample of 13,875 households in malarious areas between 2,000m and 2,500m above sea level. The goal was to ensure that the survey participants were representative of malaria endemic regions throughout the country. After completion of the main goals of the Multiple Indicator Survey, the remaining 7,077 DBS were made available for serological data collection for this study.

Lines 88-93 have been updated to: “Ethiopian samples were collected as part of the 2015 Ethiopia Malaria Indicator Survey (MIS) (19, 20), a national household survey that occurred between September 30 and December 10, 2015, which coincided with the high malaria transmission season and covered 13,875 households in malarious areas between 2,000m and 2,500m above sea level. Two-stage cluster randomized sampling was used to select 555 areas and 25 households within each area to ensure that the survey participants were representative of malaria endemic regions throughout the country.”

4. Explain - From the Costa Rica the youngest participant was 12 years old however, in Figure 2b, there are data for children under 12 years old.

The authors thank the reviewer for suggesting additional clarification in figure 2. In panel A of Figure 2, seropositivity to each antigen is modeled down to age 0 using data collected from Costa Rican participants ages 12 and up. In panel B showing the LOESS regression curves, the x-axis starts at 12 years and the curve stops at 90 years. We have revised the legend for panel B to indicate the x-axis start to avoid confusion: “The x-axis for the LOESS regression curves starts at age 12, corresponding to the youngest participant from the Costa Rican survey.”

5. In the discussion, in relation to Ethiopia, it would be recommended comparing the data from the different selected areas of the country relating to seroconversion rate and the prevalence of the different species of plasmodium. Use tables and figures to relate this data.

The authors thank the reviewer for this interesting question. Based on the Ethiopian Public Health Institute’s National Strategic Plan 2021-2025, the most complete and recent aggregated regional data, the P. falciparum/P. vivax ratio has maintained at 70/30% for these parasite species for most regions of the country. Even with changes over time, P. falciparum has remained the dominant malaria species in Ethiopia. Specifically with serological data, this study examined population exposure over long periods of time due to the longevity of IgG responses. Based on these factors, it would be difficult to correlate temporal differences throughout the country with serological data.

---

## [Decision Letter · Decision Letter 1]

18 Apr 2022

PONE-D-22-01720R1The use of a chimeric antigen for Plasmodium falciparum and P. vivax seroprevalence estimates from community surveys in Ethiopia and Costa RicaPLOS ONE

Dear Dr. McCaffery,

Thank you for submitting your manuscript for review to PLoS ONE. After careful consideration, we feel that your manuscript will likely be suitable for publication if the authors revise it to address critical points raised  now by the reviewer. According to reviewer, there are some specific areas where further improvements would be of substantial benefit to the readers.

We look forward to receiving your revised manuscript.

Kind regards,

Luzia Helena Carvalho, Ph.D.

Academic Editor

PLOS ONE

Journal Requirements:

Reviewers' comments:

Reviewer's Responses to Questions

**Comments to the Author**

1. If the authors have adequately addressed your comments raised in a previous round of review and you feel that this manuscript is now acceptable for publication, you may indicate that here to bypass the “Comments to the Author” section, enter your conflict of interest statement in the “Confidential to Editor” section, and submit your "Accept" recommendation.

Reviewer #1: All comments have been addressed

Reviewer #2: (No Response)

2. Is the manuscript technically sound, and do the data support the conclusions?

Reviewer #1: Yes

Reviewer #2: Yes

3. Has the statistical analysis been performed appropriately and rigorously? 

Reviewer #1: Yes

Reviewer #2: Yes

4. Have the authors made all data underlying the findings in their manuscript fully available?

Reviewer #1: Yes

Reviewer #2: Yes

5. Is the manuscript presented in an intelligible fashion and written in standard English?

Reviewer #1: Yes

Reviewer #2: Yes

6. Review Comments to the Author

Reviewer #1: The authors carefully answered the questions raised above.

The modifications contributed to a better understanding of the study.

The research described in this article is unprecedented and highly relevant.

Reviewer #2: Introduction:

Line 69: Sorry if I misunderstood, but reference 15 (PMID: 27708348) doesn’t show results about U.S travelers, only 16 (PMID: 33579292). Additionally, you should mention the results of a seroepidemiological study involving individuals naturally exposed to P. vivax and cross-reactivity in individuals with active infection by P. falciparum from 16 (Figure 6). Moreover, based on which results did the authors decide to use a P. vivax chimeric protein to evaluate P. falciparum antibodies? It is important to justify. I felt confused reading the introduction.

Material and methods

86-97: The authors should include the number of individuals enrolled in the study Costa Rica = 7,077 ?

127-134: To clarify. The protocol was performed overnight. The mix reagent containing conjugated antibodies (anti-Human IgG) and samples were added to the well. Did the reaction happen with all components (primary, secondary, and Streptavidin-PE) add at the same time? Why did the authors include IgG4 in to mix reaction?

138: MFI signal was generated for a target of 50 beads/region = 50 beads per antigen?

Results

153-156: The authors should describe data from Table1 according to table data. First, Costa Rica and, than Ethiopia.

162-164: I suggest the authors a table with seroprevalence data according to the region to make easy the reading of data.

194-205: NANP6 or NANPx5 peptide?

Discussion

241-247: To clarify, the NANP sequence was used as a tag to improve the purification o protein (NANP-6, was included at the C-terminus for biochemical characterization of antigenic integrity and to provide an optimal affinity purification tag, ref 15). In previously published works (refs 15 and 16), a cross-reactivity with P. falciparum was described. However, it was demonstrated that this reactivity is not due to the presence of the PfCSP NANP repeat region alone. How do the authors justify the importance of this sequence on the reactivity of antibodies in this work? I felt confused When I read previous data.

7. PLOS authors have the option to publish the peer review history of their article (what does this mean?). If published, this will include your full peer review and any attached files.

Reviewer #1: No

Reviewer #2: **Yes: **Jéssica Rafaela dos Santos Alves

---

## [Author Response · Author response to Decision Letter 1]

19 Apr 2022

Response to Reviewer Comments

Manuscript Title: The use of a chimeric antigen for Plasmodium falciparum and P. vivax seroprevalence estimates from community surveys in Ethiopia and Costa Rica

PLOS ONE Submission ID: PONE-D-22-01720

Review Comments to the Author:

Reviewer #1: The authors carefully answered the questions raised above.

The modifications contributed to a better understanding of the study.

The research described in this article is unprecedented and highly relevant.

The authors thank the reviewer for their positive feedback of our manuscript.

Reviewer #2: 

1. Introduction: 

a. Line 69: Sorry if I misunderstood, but reference 15 (PMID: 27708348) doesn’t show results about U.S travelers, only 16 (PMID: 33579292). 

The authors apologize for the confusion. Original citation 15 is the original production of the chimeric protein as a vaccine candidate and includes the relevant details regarding design. Original citation 16 is the results of the serology studies from malaria infected US travelers. 

We have revised lines 68-72 to include these citations in their appropriate locations.

b. Additionally, you should mention the results of a seroepidemiological study involving individuals naturally exposed to P. vivax and cross-reactivity in individuals with active infection by P. falciparum from 16 (Figure 6). 

The authors thank the reviewer for this suggestion. Lines 70-74 of the introduction currently state “It was evident from this study that this chimeric antigen captured IgG from a majority of returning U.S. travelers with PCR confirmed malaria infection regardless of the Plasmodium species responsible for infection (15). Furthermore, an increased assay signal was observed for PvRMC-MSP1 compared to recombinant PvMSP1 in 34 out of 38 active P. vivax infections (15).”

c. Moreover, based on which results did the authors decide to use a P. vivax chimeric protein to evaluate P. falciparum antibodies? It is important to justify. I felt confused reading the introduction.

Thank you for pointing out that the rationale of this study is not sufficiently clear. Currently the rationale of this study is summarized on lines 65-67: “It remains to be determined if chimeric antigenic constructs from different malaria parasite species can be used to collect serology data for multiple species in serosurveys,” and lines 75-82: “A better understanding of the utility of chimeric antigens for capturing antibody responses in endemic populations with different levels of malaria exposure will help determine future applications of such tools for malaria serological studies. Therefore, this study evaluated the ability of PvRMC-MSP1 to capture IgG from dried blood spot samples (DBS) collected from Ethiopia and Costa Rica. Ethiopia was selected for this study because the country is co-endemic for both P. falciparum and P. vivax, making up 69% and 27% of malaria cases, respectively (17). Costa Rica was selected as a representative low, mono-species Plasmodium endemic region, and as of 2017, 98% of the country was considered malaria-free, with only three remaining foci of active transmission for P. vivax (18).”

It is important to note that we are not suggesting that all anti-P. falciparum antibodies are captured by this chimeric antigen since it is based on P. vivax MSP1 and the NANP-repeat region of P. falciparum CSP. Due to the CSP region of the chimeric protein, as well as the promiscuous B cell epitopes from P. vivax MSP1, we have the capacity to capture some amount of anti-P. falciparum IgG. We want to emphasize in this publication is that there currently isn’t an ideal antigen that can capture all IgG responses induced following exposure to the different malaria species. To address this gap, we present a proof of concept that an engineered antigen can be used to capture antibody responses from exposure to different malaria species, and that this is a useful approach to consider when developing future engineered antigens. We believe that this idea is sufficiently covered in the discussion. 

Regarding the contribution of the P. vivax MSP1 portion of the chimera versus the P. falciparum CSP NANP repeat region to contribute to the capture of IgG from P. falciparum-exposed persons, as part of our previous publication where we investigated the ability of US travelers with active malaria infection to recognize PvRMC-MSP1 as compared to recombinant MSP1 proteins from the four main malaria species, we included in figure 7 a comparison of the PfCSP signal to the PvRMC-MSP1 signal for the 181 individuals with P. falciparum infection. 

Overall, we found “many samples were double-positive, responding to both of these antigens (Fig. 7). Additionally, some plasma samples showed a correlation of assay signals between the two antigens, tracking on a y = x reference line. However, some of these assay signals from P. falciparum infections were non-existent for PfCSP yet showed very high PvRMC-MSP1 IgG binding. No samples were IgG positive to PfCSP alone.” 

We interpreted the double positive signals as normal for a population with P. falciparum exposure and did not conclude that PfCSP was responsible for the signals observed from P. falciparum patients as “some of these assay signals from P. falciparum infections were non-existent for PfCSP yet showed very high PvRMC-MSP1 IgG binding. No samples were IgG positive to PfCSP alone.” 

2. Material and methods

a. 86-97: The authors should include the number of individuals enrolled in the study Costa Rica = 7,077 ?

In lines 165-166 of the results, we state “From the 2015 Ethiopia MIS, 7,077 DBS were available for serological data collection, and 851 DBS were available from the 2015 Costa Rica survey.”.

On line 107 of the methods, we state “In the Costa Rican canton of Matina in 2015, a total of 851 individuals were enrolled in their households and provided blood samples for DBS creation.”

For improved clarity, we have also added this statement to line 106-106: “From the 2015 Ethiopia MIS, 7,077 DBS were available for serological data collection.”

b. 127-134: To clarify. The protocol was performed overnight. The mix reagent containing conjugated antibodies (anti-Human IgG) and samples were added to the well. Did the reaction happen with all components (primary, secondary, and Streptavidin-PE) add at the same time? Why did the authors include IgG4 in to mix reaction?

The reviewer is correct, the binding of beads to hIgG, hIgG to biotinylated detection Ab, and binding of biotinylated Ab to streptavidin-PE all happens simultaneously when these reagents are mixed together. 

The supplementary anti-IgG4-BIOT is added as the anti-hIgG included in the formulation from Southern Biotech does not pick up hIgG4 very well. This was noted from previous experiments and published in 2020 in the AJTMH by Jeffrey W. Priest et al. To explain this better in the manuscript text, we have updated lines 145-147 to include “. Anti-IgG4 was added separately to the anti-human IgG mixture as we have previously found that anti-IgG4 responses are difficult to determine using only the standard anti-IgG cocktail,” and cited this publication. 

c. 138: MFI signal was generated for a target of 50 beads/region = 50 beads per antigen?

The reviewer is correct that a minimum of 50 beads per region (fluorescent signal) are required by the Luminex MagPix to produce signal, as stated in line 149 “MFI signal was generated for a target of 50 beads/region.”. Region in this case is the pre-gated bead region, or fluorescence signal from each bead type used in the assay. We separately conjugate each fluorescently tagged bead obtained from the manufacturer to one antigen of interest to allow us to obtain unique signals for PvRMC-MSP1 vs recombinant PvMSP1 vs recombinant PfMSP1. The multiplex platform allows us to obtain the level of antibody recognizing each antibody simultaneously in one well.

3. Results

a. 153-156: The authors should describe data from Table1 according to table data. First, Costa Rica and, than Ethiopia.

As suggested, we have revised table 1 to include the demographic data from Ethiopia first, above that of Costa Rica, to keep discussion of Ethiopia and then Costa Rica consistent throughout the manuscript.

b. 162-164: I suggest the authors a table with seroprevalence data according to the region to make easy the reading of data.

We thank the reviewer for this interesting question. It is possible to plot and compare the seroprevalence of antibodies to each of the three MSP1 antigens tested by region within Ethiopia, but we believe that this is outside of the scope of this study. Instead, we want to emphasize the seroprevalence to each antigen in the population as a whole, especially since Costa Rica is approaching elimination and our samples were obtained from only one region of Costa Rica.

c. 194-205: NANP6 or NANPx5 peptide?

The reviewer is correct that this passage is somewhat confusing. We have revised accordingly.

The PvRMC-MSP1 protein contains six copies of the NANP repeat peptide from the P. falciparum circumsporozoite protein, abbreviated as (NANP)6 on lines 124-125: “the (NANP)6 repeat peptide from P. falciparum circumsporozoite protein (CSP)”. This is also indicated on lines 211-214 “Although PvRMC-MSP1 is based primarily on the P. vivax Belem sequence, six copies of the NANP repeat sequence from the P. falciparum circumsporozoite protein (PfCSP) are present at the C-terminus of PvRMC-MSP1 and were included during the design of PvRMC-MSP1 to serve as an additional purification tag.” 

Additionally, we have revised lines 214-216 to state “To determine if antibody recognition and binding to the PfCSP region is responsible in part for the assay signals observed for PvRMC-MSP1, the MFI-bg signals were compared between the chimera and a peptide containing five copies of the PfCSP NANP peptide (NANPx5, S1 Figure).”

4. Discussion

a. 241-247: To clarify, the NANP sequence was used as a tag to improve the purification o protein (NANP-6, was included at the C-terminus for biochemical characterization of antigenic integrity and to provide an optimal affinity purification tag, ref 15). 

In previously published works (refs 15 and 16), a cross-reactivity with P. falciparum was described. However, it was demonstrated that this reactivity is not due to the presence of the PfCSP NANP repeat region alone. How do the authors justify the importance of this sequence on the reactivity of antibodies in this work? I felt confused When I read previous data.

Thank you for taking the time to look into our relevant previous publications as part of your review of this current manuscript. 

The reviewer is correct that the NANP sequence was initially included in the PvRMC-MSP1 protein as a purification tag during its design as a vaccine candidate. In the publication where we investigated the ability of US travelers with active malaria infection to recognize PvRMC-MSP1 as compared to recombinant MSP1 proteins from the four main malaria species, we included in figure 7 a comparison of the PfCSP signal to the PvRMC-MSP1 signal for the 181 individuals with P. falciparum infection. 

Overall, we found “many samples were double-positive, responding to both of these antigens (Fig. 7). Additionally, some plasma samples showed a correlation of assay signals between the two antigens, tracking on a y = x reference line. However, some of these assay signals from P. falciparum infections were non-existent for PfCSP yet showed very high PvRMC-MSP1 IgG binding. No samples were IgG positive alone to PfCSP.” 

We interpreted the double positive signals as normal for a population with P. falciparum exposure and did not conclude that PfCSP was responsible for the signals observed from P. falciparum patients as “some of these assay signals from P. falciparum infections were non-existent for PfCSP yet showed very high PvRMC-MSP1 IgG binding. No samples were IgG positive alone to PfCSP.”

---

## [Decision Letter · Decision Letter 2]

12 May 2022

The use of a chimeric antigen for Plasmodium falciparum and P. vivax seroprevalence estimates from community surveys in Ethiopia and Costa Rica

PONE-D-22-01720R2

Dear Dr. McCaffery,

We’re pleased to inform you that your manuscript has been judged scientifically suitable for publication and will be formally accepted for publication once it meets all outstanding technical requirements.

Kind regards,

Luzia Helena Carvalho, Ph.D.

Academic Editor

PLOS ONE

Additional Editor Comments (optional):

Reviewers' comments:

Reviewer's Responses to Questions

**Comments to the Author**

1. If the authors have adequately addressed your comments raised in a previous round of review and you feel that this manuscript is now acceptable for publication, you may indicate that here to bypass the “Comments to the Author” section, enter your conflict of interest statement in the “Confidential to Editor” section, and submit your "Accept" recommendation.

Reviewer #1: All comments have been addressed

Reviewer #2: All comments have been addressed

2. Is the manuscript technically sound, and do the data support the conclusions?

Reviewer #1: Yes

Reviewer #2: Yes

3. Has the statistical analysis been performed appropriately and rigorously? 

Reviewer #1: Yes

Reviewer #2: Yes

4. Have the authors made all data underlying the findings in their manuscript fully available?

Reviewer #1: Yes

Reviewer #2: Yes

5. Is the manuscript presented in an intelligible fashion and written in standard English?

Reviewer #1: Yes

Reviewer #2: Yes

6. Review Comments to the Author

Reviewer #1: The authors corrected the suggested changes. And all the parts that had generated doubts were clarified.

Reviewer #2: (No Response)

7. PLOS authors have the option to publish the peer review history of their article (what does this mean?). If published, this will include your full peer review and any attached files.

Reviewer #1: No

Reviewer #2: No

---

## [Editor Report · Acceptance letter]

16 May 2022

PONE-D-22-01720R2 

The use of a chimeric antigen for *Plasmodium falciparum* and *P. vivax* seroprevalence estimates from community surveys in Ethiopia and Costa Rica 

Dear Dr. McCaffery:

I'm pleased to inform you that your manuscript has been deemed suitable for publication in PLOS ONE. Congratulations! Your manuscript is now with our production department. 

Kind regards, 

on behalf of

Dr. Luzia Helena Carvalho 

Academic Editor

PLOS ONE